# Synthesis and Application of Hybrid Aluminum Dialkylphosphinates as Highly Efficient Flame Retardants for Polyamides

**DOI:** 10.3390/polym15234612

**Published:** 2023-12-04

**Authors:** Qiang Yao, Weihong Cao, Yueying Zhao, Tianbo Tang

**Affiliations:** Key Laboratory of Bio-Based Polymeric Materials Technology and Application of Zhejiang Province, Ningbo Institute of Materials Technology and Engineering, Chinese Academy of Sciences, Ningbo 315201, China; caoweihong@nimte.ac.cn (W.C.); zhaoyueying@nimte.ac.cn (Y.Z.); tangtianbo@nimte.ac.cn (T.T.)

**Keywords:** hybrid phosphinate, flame retardant, polyamide, vaporization, char

## Abstract

Hybrid aluminum dialkylphosphinates were synthesized from mixed diethyl-, ethylisobutyl-, and diisobutylphosphinates and Al^3+^ in water. The XRD, DSC, and TGA results of these Al phosphinates established that phosphinate ligands are randomly distributed in the species. The thermal and thermoxidative stabilities of the hybrid phosphinates were easily adjustable by varying the ratio of phosphinate ligands, a desirable feature for efficient flame retardants. The hybrid aluminum dialkylphosphinates with a relatively low ratio of diethylphosphinate demonstrated higher efficiency than Al diethylphosphinate and Al diisobutylphosphinate in flame-retarding polyamide 66. Detailed investigations on the thermal and thermoxidative stabilities of Al dialkylphosphinates and the morphologies of char obtained in UL-94 tests revealed that timely vaporization of degradation products of hybrid dialkylphosphinates at a temperature which closely matches the degradation temperature of polyamides and their ability to promote char formation of polyamides are two key factors which contribute to the excellent performance of hybrid aluminum dialkylphosphinates.

## 1. Introduction

Metal phosphinates are one of the most important halogen-free flame retardants for polymeric materials and have found wide applications in engineering resins such as polyamides and polyesters. In particular, aluminum hypophosphite (AP) [1,2,3] and aluminum diethylphosphinate (ADP) [4,5,6] have achieved great commercial success as eco-friendly flame retardants. However, AP suffers from low thermal stability with a tendency to produce toxic phosphine gas, while ADP itself does not work well and needs synergists in industrially important aliphatic polyamides [7,8,9]. On the other hand, aluminum diisobutylphosphinate (ABP) possesses reasonably good thermal stability and exhibits a surprisingly high efficiency in polyamides, despite its low phosphorus content [10]. However, the synthesis of ABP in the greenest solvent, i.e., water, has been proven difficult and much more costly than that of ADP [11]. Thus, a new type of halogen-free flame retardant with good thermal stability, high efficiency, and production economy is extremely desirable for polyamides.

To achieve this goal, we have pursued reconstruction of alkyl substituents on the phosphorus atom on the basis of the mode of action of metal phosphinates. It has been shown that AP mainly works in the condensed phase with its aptitude to promote char formation of polyamides, while ADP primarily acts in the vapor phase with its flame inhibition [1,5]. For ABP, its impressively high efficiency can be traced back to its easy vaporization at a relatively low temperature which occurs well before thermal degradation of polyamides, but weak condensed phase action has also been noted [12,13]. In view of these results, it is logical to presume that an ability to promote charring of polyamides and timely vaporization of flame retardants are two key factors for metal dialkylphosphinates to work well in polymers.

For timely vaporization, metal dialkylphosphinates must have a bulky group chemically attached to the phosphorus atom, since the bulky group can reduce the metal ion’s expose to polar neighboring P=O attractions and facilitate the evaporation of metal dialkylphosphinates. On the other hand, charring demands chemical interaction between polymers and flame retardants or their degradation products. Given that AP substantially enhances char formation of polyamides, we speculated that dialkylphosphinates which can generate hypophosphite via the elimination of alkene at an elevated temperature, a reverse reaction of P–H addition to alkene, should possess good condensed phase action. Consequently, an isobutyl group has been chosen to be incorporated into dialkylphosphinates, since its elimination to isobutylene and P–H is favorable at an elevated temperature.

In this paper, we report the facile synthesis of hybrid aluminum (III) with mixed diethyl-, ethyl isobutyl-, and diisobutylphosphinates in water and their applications as efficient flame retardants for polyamides, particularly polyamide 66. These hybrid phosphinates are intrinsically different from the physical mixture of pure Al dialkylphosphinates in that the former has structurally distinct dialkylphosphinate ligands to coordinate to the same Al^3+^. Metal phosphinates with mixed hybrid ligands have occasionally been mentioned in the literature but have not been suggested as a flame retardant for polymers [14,15,16,17]; however, due to their structural versatility and high efficiencies demonstrated in our experiments, hybrid metal dialkylphosphinates will certainly open up plenty of opportunities in the development of new flame retardants and flame-retardant polymers.

## 2. Materials and Methods

### 2.1. Materials

Sodium hypophosphite, sodium persulfate, and aluminum sulfate octahydrate were purchased from Sinopharm Chemical Reagent Co., Ltd. (Shanghai, China). Ethylene and isobutylene were bought from Shanghai Wetry Standard Gas Analysis Technology Co., Ltd. (Shanghai, China). Antioxidant 168 was acquired from Strem Chemicals; Antioxidant 1010 was obtained from Shanghai Macklin (Shanghai, China). ADP (Exolit^®^ OP1230) was from Clariant (Muttenz, Switzerland). ABP with a Mp = 293 °C was prepared according to the literature [11]. Polyamide 6 (PA6, Zytel 73G30L NC010) and polyamide 66 (PA66, Zytel 70G33L NC010) were produced by Dupont (Wilmington, DE, USA). All materials were used without further purification.

### 2.2. Synthesis of Hybrid Al Phosphinates with Mixed Ligands

In a typical procedure, 100 g sodium hypophosphite (0.943 mol) and 500 g water were charged into a 1 L stainless steel reactor equipped with a mechanical stirrer, a gas inlet, an initiator inlet, a pressure gauge, and a safety rupture disc. The reactor was first purged with nitrogen and then degassed via vacuum. Isobutylene from a pressure cylinder was introduced carefully. The reaction medium was heated to 90 °C, and an aqueous solution of sodium persulfate (4 wt.%) was evenly pumped in at a rate of 10 mL/h. The reaction pressure was kept at 0.3 MPa. After a desired ratio of isobutylphosphinate was obtained, which was determined with ^31^P NMR, the flow of isobutylene was stopped and ethylene was instead introduced until the end of the reaction. During the ethylene reaction, the pressure was maintained at 0.8 MPa. The total reaction time varied from 10 to 25 h depending on the ratio of isobutylene.

The above solution was slowly mixed with 20 wt.% aqueous aluminum sulfate octahydrate, immediately generating plenty of white precipitate at 70 °C. The precipitate was first filtered, then washed with water and dried at 120 °C. The compositions of the products were determined with ^31^P NMR in an aqueous NaOH solution. Table 1 shows the mole ratios of diethyl-, ethylisobutyl-, and diisobutylphosphinates, phosphorus content (P%, actual values were determined using ICP), and median particle size (D_50_) in each product.

### 2.3. Preparation of Polyamides/PFR Blends

Polyamide 66 containing 33 wt.% glass fiber was dried at 80 °C for approximately 4 h before use. Flame-retardant polyamides comprising 12.5–20 wt.% of Al phosphinates reported in Table 1 were prepared via melt compounding at 260–280 °C in a Brabender mixer at a roller speed of 50 rpm. The mixing time was 5 min. After that, the blends were transferred into a mold which had been preheated at 280 °C, then pressed at 10 MPa for 5 min, followed by pressing at room temperature for 10 min. The sample plaques obtained were cut into specific dimensions and stored for further tests.

### 2.4. Characterization

NMR: ^31^P NMR analyses of the Al phosphinates in Table 1 were performed with a Bruker 400 AVANCE spectrometer (Bruker, Karlsruhe, Germany) in an aqueous NaOH solution. The ^31^P NMR measurements were run at a frequency of 162 MHz.

P%: Phosphorus content of Al phosphinates was determined using an ICP optical emission spectrometer (Perkin-Elmer, Woodbridge, ON, Canada). The samples were firstly digested at 130 °C using an aqua regia 37% HCl/70% HNO_3_ (3:1) mixture (6 mL per 0.10 g of sample) in a microwave system for 2 h. The suspensions were then filtered and diluted to 100 mL with HNO_3_ for analysis. P (213.617, 214.914) was used as the calibration standard.

Particle size: D_50_ of Al phosphinates was measured on dried samples using Heloise-oasis HELOS (SympatecGmbH, Clausthal-Zellerfeld, Germany).

XRD: X-ray powder photographs of the Al phosphinates were recorded with a Bruker AXS D8 Discover X-ray diffractomer (Bruker, Karlsruhe, Germany) in a 2θ range from 3 to 40° using Cu Kα radiation at 40 KV and 40 mA.

DSC: Differential scanning calorimeter of the Al phosphinates was performed with a Mettler Toledo DSC3+ differential scanning calorimetry (Mettler Toledo, Zurich, Switzerland) under nitrogen from 50 to 400 °C. Sample weights were of the order of 5 mg. The heating rate used was 10 °C/min.

FTIR: Fourier transform infrared spectra (FTIR) of the Al phosphinates were recorded with an Agilent Technologies Cary 660 FTIR spectrometer interfaced to a Pike Technologies GladiATR (Agilent Technologies, Santa Clara, CA, USA) with a diamond crystal at 4 cm^−1^ resolution.

TGA: Thermogravimetric analysis (TGA) experiments of the Al phosphinates and flame-retardant blends were all performed on a TA TGA 50 Analyzer (TA Instruments, Newcastle, NSW, USA). An amount of 3~5 mg of samples were chosen so the effect of quantities on the char was insignificant as verified experimentally. The samples were heated from 50 °C to 600 °C in a nitrogen or air atmosphere (60 mL/min) at a heating rate of 10 °C/min.

UL-94 vertical burning test: UL-94 vertical burning tests of flame-retardant blends were conducted with a 5400 vertical burning tester (Suzhou YangYi Vouch Testing Technology Co., Ltd., Suzhou, China) with sample dimensions of 130 mm × 13 mm × 1.6 mm according to ASTM D3801 [18].

SEM: Scanning electron microscopy (SEM) on char obtained after UL-94 tests was performed using a Hitachi Regulus SU8230 scanning electron microscope (Hitachi, Tokyo, Japan). All samples for SEM were sputtered with a thin layer of platinum before examination.

## 3. Results and Discussion

### 3.1. Characterizations of Hybrid Al Phosphinates

NMR characterization of hybrid Al with diethyl-, ethylisobutyl-, and diisobutylphosphinates was performed on their alkali solutions. Three anions, i.e., diethylphosphinate (49.67 ppm), ethylisobutylphosphinate (47.39 ppm), and diisobutylphosphinate (45.14 ppm), give their own distinctive peaks, as shown in Figure 1. This allows a quick determination of the mole fraction of each anion in the Al phosphinates. The peak at 48.57 ppm arises from the absorption of ethylbutylphosphinate which is generated as a by-product due to the oligomerization of ethylene. Table 1 lists the compositions determined from the NMR analysis.

The FT-IR spectra of the Al phosphinates are shown in Figure 2. All of the phosphinates have strong absorptions of PO_2_^−^ at 1149 and 1077 cm^−1^. These two frequencies are associated with the asymmetric and symmetric PO_2_^−^ stretching bands and vary little for all of them. The medium separation of 72 cm^−1^ between v_as_(PO_2_^−^) and v_sym_(PO_2_^−^) and the high intensities of the peaks suggest the dominant presence of symmetrical phosphinate bridging in all of the species, which is also the most common coordination mode of phosphinate ligands [19,20]. However, further examination reveals that there are additional peaks in the PO_2_^−^ stretching region, particularly for ABP and the hybrid Al phosphinates. Although only a small peak at 1046 cm^−1^ can be found for ADP, there are two sizable peaks at 1106 cm^−1^ and 1058 cm^−1^ for ABP. The large difference in the intensities of these peaks implies that unlike PO_2_^−^ in ADP, those in ABP and the hybrid Al phosphinates may have other different bonding modes, resulting from the presence of a more steric hindering isobutyl group.

Interestingly, although mixed ligands were used to synthesize hybrid Al phosphinates, the products are clearly crystalline solids, as evidenced by their XRD patterns in Figure 3. For Al phosphinates with x, y, and z < 0.7, there is only one peak present in the region between the most intense peak of ABP and that of ADP. Considering the good symmetry and relative sharpness of the peak, the crystallinity of the Al phosphinates remains high. This is consistent with an early report that a replacement of a dialkylphosphinate ligand by another of the same type does not result in a marked reduction in crystallinity since the backbone structure (-O-P-O-Metal-) of hybrid phosphinates is the most dominant factor in determining the crystallinity [17].

Additionally, it is noted that the most intense peak of hybrid Al phosphinates shifts to smaller angles with a decreasing value of x in a continuous manner. In species with x, y, and z < 0.7, no matching peaks of ABP or ADP can be found. These results support that all phosphinate ligands are randomly distributed in the species. Otherwise, discrete peaks associated with different compositions should be observed as seen in E87M13B0 and E70M29B1.

To further characterize Al phosphinates, DSC thermograms were carried out, and the results are shown in Figure 4. There is a characteristic endothermic peak at 169 °C for ADP due to a solid state transition involving the side ethyl group [17]. However, this peak moves to lower temperatures in the hybrid phosphinates and simultaneously grows broad. These outcomes obviously stem from the disordering of the side alkyl groups, further supporting that phosphinate ligands are randomly distributed.

Another interesting finding in the DSC thermograms is that both E31M63B6 and E3M52B45 show fusion behaviors. The former has a melting peak at 372–392 °C and the latter at 314–322 °C. The relatively broad melting peaks suggest the imperfectness of crystals. However, the heat of fusion in E3M52B45, as measured from its DSC thermogram, is minimally 161.76 J/g depending on the start point and the end temperature. This value reaches more than 80% of that of pure Al diisobutylphosphinate which has a sharp melting point at 293 °C and a heat of fusion of 201.87 J/g. This is apparently in line with the XRD results that show the length and order of the side alkyl groups do not significantly change the crystallinity of hybrid phosphinates.

### 3.2. Thermal Stability of Al Phosphinates and Flame-Retardant Polyamides

Figure 5 shows the thermogravimetric analysis of the Al phosphinates under nitrogen, and the results are listed in Table 1. Evidently, ABP is the first one to lose mass, while ADP possesses the highest thermal stability as revealed by T_5%_, the temperature at 5% mass loss. For the other Al phosphinates, their thermal stability decreases with a reducing value of x. Since metal dialkylphosphinates are typically coordinate compounds which are connected via the dative bond between Al^3+^ and P=O [14,21], thermal stability of the Al phosphinates is undoubtedly dependent on the strength of the dative bond which is, in turn, decided by the distance between the donor (P=O) and the acceptor (Al^3+^). A bulky substituent on phosphorus would increase the separation of Al^3+^ and P=O and result in a weak dative bond which reduces the symmetry of PO_2_^−^ or changes the geometry of the metal center. As a matter of fact, this can be the reason for ABP to have large extra PO_2_-stretching frequencies as seen in its FTIR spectrum. Consequently, thermal stability of the hybrid phosphinates is governed by the amount of isobutyl group in their structures. The more isobutyl group the hybrid phosphinate has, the lower thermal stability it has. This trend is well correlated to the values of d-spacing which are inversely proportional to sinθ in the XRD measurement.

Besides the curves of thermal stability, the shape of the TGA derivative curves of the Al phosphinates can also provide insight on their structures. For example, E87M13B0 shows a peak at 443 °C which is clearly associated with degradation of the ADP component, consistent with its XRD pattern. However, no ADP peak can be found in the TGA derivative curve of E70M29B1, even though there is a small amount of ADP, as indicated in the XRD pattern. The missing ADP peak implies its accelerated thermal degradation. Since the breakdown of ADP is mainly through cyclization to dimeric species [22], the early fragmentation of the ADP component suggests that there exists a weakened dative bond between Al^3+^ and P=O, which, in turn, implies the doping of the ethylisobutylphosphinate ligand in the crystal lattice of ADP. Likewise, further doping of ligands should lead to coalescence of peaks in the derivative curves due to the increased amount of weakened dative bonds. In fact, E3B52B45 predominantly shows one peak, supporting a random doping of phosphinate ligands in this species.

In addition, the thermogravimetric analysis of flame-retarded PA66 under nitrogen has also been examined in detail. Figure 6 shows its TGA and DTG curves, and Table 2 lists the results. It can be readily concluded that the incorporation of Al phosphinates accelerates thermal degradation of PA66, as evidenced by its reduced T_5%_ and Tmax, the temperature at the maximum rate of mass loss of flame-retarded polyamides. However, unlike T_5%_ which shows a unidirectional decrease, Tmax firstly declines, reaches a minimum value in E28M68B4, and then bounces back to higher temperatures in E3M52B45 and ABP. For the latter two phosphinates, they degrade and vaporize so early that their chemical interaction with PA66 is significantly reduced. Thus, the values of T_max_ of flame-retarded PA66s increase again.

On the other hand, in terms of char yields under nitrogen, the actual values agree well with the calculated ones, as seen in Table 2. Thus, although Al phosphinates chemically interact with PA66 during thermal degradation of the latter, the main chemistry leading to char formation is probably not changed. This is likely because Al phosphinates only catalyze the initial intramolecular cyclization of PA66 [23]. After that, Al phosphinates and the intermediates generated from the initial thermal degradation of PA66 do not chemically interact, or their interactions are not favorable for char formation.

### 3.3. Thermoxidative Stability of Al Phosphinates and Flame-Retardant Polyamides

Thermoxidative stability of Al phosphinates and flame-retardant PA66 is also investigated to gain insight into the role of oxygen. Figure 7 shows the TGA curves of the Al phosphinates under air atmosphere. It is noted that the TGA curves obtained under nitrogen and air are nearly superimposable for the same Al phosphinate during the initial degradation stage. In fact, the plot of T_5%_ under air vs. those under nitrogen shows a nearly straight line, as illustrated in Figure 8, so oxidation of the side alkyl groups is not involved in the initial degradation. Instead, it is the inorganic main chain (-O-P-O-metal-) that serves as the weak point, likely the dative bond, to start the decomposition. However, as the cleavage of the main chain takes place and the rigid crystal lattice collapses, oxidation must occur to the alkyl groups because all of the Al phosphinates produce a significant amount of residues except ABP, as shown in Table 1. The higher ratio of diethylphosphinate there is in the species, the larger the amount of residue it produces. For ABP, it completely vaporizes before it undergoes oxidation, so almost no residue can be found.

On the other hand, it is surprisingly noticed that T_5%_ of FR-PA66 under air is higher than that obtained under nitrogen in spite of the fact that the stability of PA66 itself is lower under air than under nitrogen. The increased stability of FR-PA66 in the presence of oxygen implies that oxygen reduces the deleterious effect of Al phosphinates on thermal degradation of PA66, and it might be attributed to the competing reactions between oxidation and the intramolecular cyclization of PA66. Since oxidation can occur on the methylene group adjacent to carbonyl of the amide group, which is also involved in the thermal degradation of PA66 [24], the effect of Al phosphinates on the intramolecular breakdown of PA66 is reduced. As a result, a smaller difference between Tmax of PA66 and Tmax of FR-PA66 is attained under air than under nitrogen. However, catalyst poisoning of phosphinates by the oxidation products of PA66 cannot be ruled out.

What is more, the effect of oxygen is not limited to the stability of FR-PA66; the presence of oxygen also facilitates the formation of char. In contrary to a zero increase in char yields under nitrogen, FR-PA66s produce more than a theoretical amount of char under air, as can be seen in Table 2. A thorough analysis establishes that the difference between the actual values and the calculated ones firstly increases then decreases with a reducing value of x, as illustrated in Figure 9. This trend is very similar to the changes in the Tmax of FR-PA66s, and hence, can be attributed to the same factors, i.e., ADP is so thermoxidatively stable that it only marginally interacts with the charring process of PA66, while ABP is too volatile to influence the char formation of PA66. Thus, neither ADP nor ABP is able to efficiently enhance the char formation of PA66. On the contrary, the hybrid Al phosphinates that degrade at a right temperature which closely matches the charring temperature of PA66 can effectively promote the char formation of the latter.

To probe species accounting for enhanced char formation in FR-PA66, a mixture of E63M28B1 and ADP was subjected to heat treatment at 300 °C. The residues were examined using ^31^P NMR, and the results are shown in Figure 10. On the basis of the chemical shift and splitting patterns, ethylphosphonate and phosphoric acid (as the sodium salt) can be clearly identified. The former is assumed to be produced by the reverse reaction of P-H addition to alkenes followed by oxidation, and the latter is the product of the former’s further elimination and oxidation reactions as illustrated in Figure 1. In addition, it is noted that the ratio of E60M38B2 to ADP becomes smaller after heat treatment at 300 °C, suggesting that E60M38B2 degrades and produces phosphoric acid faster than ADP, consistent with the TGA outcomes. Thus, in view of the NMR results and the char yields, it is presumed that phosphate acids, generated from elimination and oxidation reactions of the hybrid Al phosphinates, are the actual species which chemically interact with PA66 or its degradation products in the condensed phase to enhance char formation of the latter. This conclusion is in line with the earlier work on the role of phosphate acids in the charring process of polyamides [25].

### 3.4. UL-94 Results

Table 3 lists the UL-94 test results of the flame-retarded PA66. A general trend can be readily recognized that the smaller the value of x, the higher efficiency the species has, regardless of its particle size. For example, ADP completely fails at 20%, but E87M13B0 and E60M38B2 help the polymer achieve UL-94 V1 and V0 ratings, respectively. For the Al phosphinate hybrid copolymers with x ≤ 0.7 and ABP, all of them enable PA66 to achieve UL-94 V0 ratings at the same loading levels. These results demonstrate the importance of timely vapor phase action, as ABP is only marginally involved in the charring process of PA66 [12].

However, vapor phase action alone does not always ensure good flame retardancy. At a reduced loading level of 12.5%, ABP becomes less efficient and only gives PA66 a V-1 rating. Likewise, Al phosphinates with x ≥ 0.7 also display lower performance. For example, neither E87M13B0 nor E70M28B1 enable PA66 to achieve a UL-94 rating. On the other hand, the hybrid Al phosphinates with x < 0.7 demonstrate equally high efficiency even at 12.5%. The variations in the performance of these different species are remarkably parallel to the discrepancies in the char yields of flame-retarded PA66 under air, suggesting that the high efficiency of the hybrid phosphinates partly stems from their ability to promote char formation of PA66, i.e., they have a significant condensed phase action, too. As a matter of fact, a thin but sturdy layer of black char was observed during the UL-94 tests for PA66 containing hybrid Al phosphinates with x < 0.7.

In addition, a comparison of the performance of E59M0B41, a physical mixture of ADP and ABP, and E60M38B2 also strongly supports the condensed phase action of the hybrid Al phosphinates. As seen in Table 2, E59M0B41 completely fails at 20%, while E60M38B2 easily imparts a V0 rating to PA66 even at 12.5%. Since the difference between these two species is that E59M0B41 does not have a considerable condensed phase action but E60M38B2 has, as judged by the latter’s ability to increase the char formation of PA66, there is a clear correlation between the efficiency of the hybrid phosphinate and its condensed phase action in the flame retardancy of PA66.

To further confirm the condensed phase action, the quality of char was also examined. Figure 11 shows the SEM pictures of the surfaces of the residues obtained after the UL-94 tests. It can be seen that the surfaces of the residues generated from PA66/ADP and PA66/ABP are coral reef-like but those generated from PA66/E63M36B1 and PA66/ E28M68B4 are compact. This discrepancy in the quality of char certainly results from the different degrees of chemical interactions between Al phosphinates and PA66. The more chemical interaction there is in the condensed phase, the more compact the surface of the residue and the higher efficiency the Al phosphinate has. Thus, in order for Al phosphinates to work efficiently in PA66, it is best for them to have both the vapor phase action and the condensed phase action.

## 4. Conclusions

Hybrid Al phosphinates as flame retardants for PA66 were synthesized from mixed diethyl-, ethylisobutyl-, and diisobutyl-phosphinate and Al^3+^ in water. The phosphinate ligands possess mostly a symmetrical μ2-bridging mode; although, there are additional coordination modes in ABP and hybrid Al phosphinates. Despite the presence of mixed phosphinates, hybrid Al phosphinates are crystalline solids, supporting that the length and order of the side alkyl group do not significantly affect crystallinity. In fact, the DSC thermogram reveals that the heat of fusion of E3M52B45 reaches more than 80% of that of ABP.

Hybrid Al phosphinates possess thermal and thermoxidative stabilities between ADP and ABP. The less diethylphosphinate there is in the species, the earlier it starts to lose mass. The Al phosphinates with x < 0.7 show higher efficiency than ADP and ABP in flame-retarding polyamide 66. This is because ADP is too thermoxidatively stable to meaningfully interact with the charring process of PA66, while ABP is so volatile that it is not able to influence the char formation of PA66. On the other hand, hybrid Al phosphinates can degrade to phosphate acid at the right temperature which closely matches the charring temperature of PA66, and hence, can effectively promote the formation of compact char. Thus, the excellent performance of the hybrid phosphinates is attributed to their timely vaporization and their ability to promote the formation of compact char in polyamides.

## Data Availability

The data used to support the findings of this study are included in the article.

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
