# Peer review of "Synthesis and Application of Hybrid Aluminum Dialkylphosphinates as Highly Efficient Flame Retardants for Polyamides"

_polymers, 2023, doi:10.3390/polym15234612_

Round 1

Reviewer 1 Report

Comments and Suggestions for Authors

The presented manuscript is good overall, but in order to enhance its quality and presentation there are few issues that authors need to address. On the page 4, in the Characterization part, lines 125-128, authors forgot to highlight the temperature region of the DSC analysis. Likewise, line 131, instead of “Thermal gravimetric” it is better to write “Thermogravimetric”. Page 6, line 206, please substitute the term “thermometric” with “Thermogravimetric”. The same on the page 7, line 237. Page 5, line 158, please correct the “antisymmetric” into “asymmetric”. Figure 4 shows DSC curves. Authors did not mention are those curves from first or second heating cycle, or did they performed cooling between two heating cycles. Since in the first heating cycle thermomechanical history of the samples is erased, it would be very interesting to see those curves, especially since the samples were prepared in this work. On the black curve of the ADP sample in the region about 250 °C one small exothermic peak is visible; is it cold crystallization? Likewise, it is evident that, for all other samples, measurement is not finished (above 400 °C), part of curves is missing. Why authors did not measure above 400 °C? In the Table 2 authors showed only T5%. However, the Tonset (extrapolated temperature of degradation) is more valuable parameter, comparable to Tmax. Page 8, lines 250-252 authors stated: “Thus, although the Al phosphinates chemically interacts with PA66 during thermal degradation of the latter, the main chemistry leading to the char formation is probably not changed.” This fact need to be verified! Figure 7 shows TGA curves of Al phosphinates under air atmosphere. Where are the corresponding DTG curves? Page 9, lines 278-290 authors are comparing T5% of the FR-PA66 under air and nitrogen. Hence, authors are advised to insert one more Fig. by which they will visually show comparison of the TG/DTG curves of one selected sample (FR-PA66) in nitrogen and air atmosphere. Can authors correlate the results of the UL-94 vertical burning tests of flame retardant blends with results of the TGA analysis?

Comments on the Quality of English Language

Needs minor revision.

Author Response

The presented manuscript is good overall, but in order to enhance its quality and presentation there are few issues that authors need to address.

On the page 4, in the Characterization part, lines 125-128, authors forgot to highlight the temperature region of the DSC analysis.

Reply: Point taken. The temperature range has been added. “Differential scanning calorimeter of the Al phosphinates was performed on a Mettler To-ledo DSC3+ differential scanning calorimetry (DSC) from 50 to 400 °C under nitrogen.”

Likewise, line 131, instead of “Thermal gravimetric” it is better to write “Thermogravimetric”. Page 6, line 206, please substitute the term “thermometric” with “Thermogravimetric”. The same on the page 7, line 237.

Reply: Point taken. Three typos have been changed. Thanks for pointing out the errors.

Page 5, line 158, please correct the “antisymmetric” into “asymmetric”.

Reply: Point taken. It has been changed to asymmetric.

Figure 4 shows DSC curves. Authors did not mention are those curves from first or second heating cycle, or did they performed cooling between two heating cycles. Since in the first heating cycle thermomechanical history of the samples is erased, it would be very interesting to see those curves, especially since the samples were prepared in this work.

Reply: All DSC curves are taken from the first heating cycle to preserve the original solid-solid transitions and the crystallinity in order to compare the newly synthesized hybrid salts with the known commercial ADP. The peak at 169℃ is a characteristic of ADP. It moves to lower temperatures in the newly synthesized salts. This fact implies that the newly synthesized Al phosphinates have a disordering of the side alkyl groups, and is used to support that phosphinate ligands are randomly distributed, i.e. the primitive information obtained in the first cycle justifies that the new salts are not mixtures of ADP with other Al phosphinates.

On the black curve of the ADP sample in the region about 250 °C one small exothermic peak is visible; is it cold crystallization? Likewise, it is evident that, for all other samples, measurement is not finished (above 400 °C), part of curves is missing. Why authors did not measure above 400 °C?

Reply: ADP is a non-meltable solid. It has two crystalline forms which can interconvert to each other. It is believed that the transition at 250 °C is due to the modification of the crystalline structure of ADP (to see Thermochimica Acta, 2013, 551, 175-183).

The most reliable and valuable information about the crystallinity and the solid-solid transitions we can extract from DSC is at low temperature ranges. Results above 400 °C will be complicated by the vaporization or sublimation of the Al phosphinates, which can lead to the loss of samples and give misleading information.  

In the Table 2 authors showed only T5%. However, the Tonset (extrapolated temperature of degradation) is more valuable parameter, comparable to Tmax.

Reply: All of these parameters give some insights although T5% is somewhat arbitrary. Nonetheless, T5% eliminates the effect of impurities such as residual water and the static electricity of samples. It is also more practicable in assessing the processing temperature than Tonset which often overestimates the stability. For Tmax, it is very useful to deduce the degree of chemical interactions between different components during the course of degradation. Tonset is rarely used to provide such information since it essentially concerns the initial degradation.

Page 8, lines 250-252 authors stated: “Thus, although the Al phosphinates chemically interacts with PA66 during thermal degradation of the latter, the main chemistry leading to the char formation is probably not changed.” This fact need to be verified!

Reply: This statement is simply inferred from the facts that the addition of Al salts lowers thermal stability of PA66 but does not change the amount of char. Lowered thermal stability of PA66 in the presence of Al phosphinates is apparently due to the latters’ effect on thermal stability of the first through their chemical interactions. On the other hand, if a significant amount of char were observed, the main chemistry leading to the char formation would have been changed, which is not true. Thus, these two facts together allow us to reasonably arrive at such speculation.

Figure 7 shows TGA curves of Al phosphinates under air atmosphere. Where are the corresponding DTG curves? Page 9, lines 278-290 authors are comparing T5% of the FR-PA66 under air and nitrogen. Hence, authors are advised to insert one more Fig. by which they will visually show comparison of the TG/DTG curves of one selected sample (FR-PA66) in nitrogen and air atmosphere. Can authors correlate the results of the UL-94 vertical burning tests of flame retardant blends with results of the TGA analysis?

Reply: For easy understanding, DTG curves are illustrated in Figs. 5 and 6. However, Figure 7 is straightforward even without the DTG curves. So for the sake of simplicity, we did not put DTG curves in Figure 7. Nonetheless, all of the important information has been listed in Table 2 and explained in detail in the sections of 3.2 and 3.3.

As pointed out in our conclusions, timely vaporization and the ability to promote the char formation of PA66 are two key factors contributing to the excellent flame retardancy of some hybrid salts. These two features can be assessed in the TGA analysis. However, the quality of char which is also critical to the efficiency of flame retardants cannot be evaluated in the TGA analysis. Instead, it can be assessed in the SEM pictures of residues. Fortunately, in our case the hybrid salts which work well also promote the formation of high quality char. So there is a good correlation between the TGA and the UL-94 test results in our paper. However, this correlation is not universal and may not be applicable to other flame retardant blends.

Reviewer 2 Report

Comments and Suggestions for Authors

This paper presents flame retardant nature of various Al phosphinates.

There are no serious problems, so it should be accepted after correcting additional explanation.

(1) In thermal analysis, the Al phosphinates usually decomposed around 350 C. Is such (low?) temperature enough for general types of fires?

(2) Dative bonds between Al3+ ion and P=O lone pairs are mentioned in many times. Is it possible to form stable and composition-constant coordination in this case? 

(3) In Figure 3, why XRD peaks indicated clear shifts only or mainly by composition differences?

That's all.

Author Response

This paper presents flame retardant nature of various Al phosphinates.

There are no serious problems, so it should be accepted after correcting additional explanation.

(1) In thermal analysis, the Al phosphinates usually decomposed around 350 C. Is such (low?) temperature enough for general types of fires?

Reply: The TGA curves do not differentiate thermal instability from vaporization or sublimation. That is to say, the mass loss can be due to either thermal degradation or vaporization/sublimation. Many Al phosphinates actually sublime and thus have higher thermal stability than they appear in the TGA curves. On the other hand, since most polymers are processed well under 350 oC, thermal stability up to 350 oC is good enough for flame retardants since they can pass the high temperature processing without degradation. Further, a desired feature of flame retardants is that they should be able to yield true flame retarding species such as phosphate acid or PO/PO2 radicals before degradation of polymers. If true flame retarding species are generated at a temperature close to or higher than the degradation temperature of polymers, it may be too late for flame retardants to work efficiently as seen in the case of ADP.

Thus, to answer this question, Al phosphinates with a mass loss beginning at 350oC is good since they can pass the high temperature processing and are able to generate true flame retarding species before degradation of polymers. 

 (2) Dative bonds between Al3+ ion and P=O lone pairs are mentioned in many times. Is it possible to form stable and composition-constant coordination in this case? 

Reply: The IR spectra indicate a symmetrical µ2-bridging mode of PO2 anion. Thus, the dative bond of (P=O)-Al is really indistinguishable from (P-O)-Al and they are equivalent in reality. As a result, coordinating ligands cannot be differentiated either.

In our paper, we use (-O-P-O-Metal-) to describe them (to see line 178). In this sense, the coordination dative bond is as stable as (P-O)-Al. However, the bond strength is affected by the steric hindering substituent on P.

On the other hand, compounds with composition-constant coordination (if it means salts with the same ligand) are not formed in Al phosphinates with x<0.7. If they were formed, the XRD spectra would have given a distinct peak for each of them as seen in the Al phosphinates with x>0.7. The absence of discrete peaks associated with different compositions implies the random distribution of ligands, which is also supported by their TGA and DSC results.

 (3) In Figure 3, why XRD peaks indicated clear shifts only or mainly by composition differences?

Reply: XRD technique can identify crystal materials based on their diffraction peaks by comparing their unique d-spacings with standard materials. In our paper, we use the known ADP as the standard sample and synthesized the new salts in a similar way (also similar to the synthesis of ADP) to eliminate factors other than composition related. Thus, the presence of sharp peaks in the newly synthesized Al phosphinates implies these materials are crystals, which allows us to use XRD to differentiate them from ADP. The large peak shift implies the different lattice parameters which are mainly caused by the dopants with different compositions and hence different sizes. For more information, please see reference 14-17 cited in this paper.

That's all.
